# Fungal-Mediated Silver Nanoparticle and Biochar Synergy against Colorectal Cancer Cells and Pathogenic Bacteria

**DOI:** 10.3390/antibiotics12030597

**Published:** 2023-03-16

**Authors:** Moath Alqaraleh, Khaled M. Khleifat, Maha N. Abu Hajleh, Husni S. Farah, Khaled Abdul-Aziz Ahmed

**Affiliations:** 1Pharmacological and Diagnostic Research Center (PDRC), Faculty of Pharmacy, Al-Ahliyya Amman University, Amman 19328, Jordan; 2Department of Medical Laboratory Sciences, Faculty of Allied Medical Sciences, Al-Ahliyya Amman University, Amman 19328, Jordan; 3Department of Medical Laboratory Sciences, Faculty of Science, Mutah University, Al-Karak 61710, Jordan; 4Department of Cosmetic Science, Pharmacological and Diagnostic Research Centre, Faculty of Allied Medical Sciences, Al-Ahliyya Amman University, Amman 19328, Jordan

**Keywords:** antitumor activity, antibacterial, silver nanoparticles, biochar, colorectal cancer

## Abstract

Background: Silver nanoparticles (AgNPs) are attractive substrates for new medicinal treatments. Biochar is pyrolyzed biomass. Its porous architecture allows it to hold and gather minuscule particles, through which nanoparticles can accumulate in its porous structure. This study examined AgNPs’ antibacterial and anticancer properties alone and combined with biochar. Methods: The fungus *Emericella dentata* was responsible for biosynthesis of AgNPs. The characterization of AgNPs using STEM images and a Zetasizer was carried out. Accordingly, the antibacterial and antiproliferation activity of AgNPs and biochar was studied using MIC and MTT assays, respectively. To evaluate the antiangiogenic and anti-inflammatory effects of AgNPs with biochar, VEGF and cytokines including TNF alpha, IL-6 and IL-beta were tested using an ELISA assay. Results: The size of the AgNPs ranged from 10 to 80 nm, with more than 70% of them being smaller than 40 nm. The combination of AgNPs and biochar enhanced the antibacterial activity against all tested bacteria. Furthermore, this combination showed antiproliferative properties against HT29 cancer cells with high selectivity to fibroblasts at low concentrations. AgNPs with biochar significantly reduced VEGF and proinflammatory cytokine expression levels. Conclusions: Biochar and AgNPs may be novel treatments for bacteria and colorectal cancer cells, according to the current findings.

## 1. Introduction

Nanotechnology is a broad multidisciplinary field that has evolved worldwide [1,2]. It provides nanoscale components with a high surface area, improving their physicochemical characteristics over their bulk counterparts [3]. In particular, metal nanoparticles and their biological activity are important areas of growing interest in the research field [4,5]. Silver nanoparticles (AgNPs) are a prominent nanoproduct due to their unique chemical, physical and biological properties. They are considered promising platforms for the development novel therapeutic agents, acting against various cancer cells and drug-resistant pathogens [6,7].

Nanotechnology-based medication has been considered a new prospect for opposing microbial drug resistance. AgNPs can effectively kill *Escherichia coli, Klebsiella pneumonia* and *Staphylococcus aureus*, fungi (such as *Candida albicans*, *Aspergillus niger*) and viruses (such as hepatitis B, respiratory syncytial infection, herpes simplex infection type 1, monkeypox infection and human immunodeficiency virus) [8,9]. Additionally, nanotechnology-based cancer therapy is a new strategy and is considered one of the most promising research directions for cancer [10]. AgNPs could induce cell death of colon cancer cells by destroying the cytoskeleton, altering membrane nanostructures and reducing cell biomechanics [11]. Likewise, Acharya et al. [12] demonstrated that biogenically synthesized AgNPs possess a selective apoptotic effect against colon cancer (HCT-116) cells through proapoptotic protein activation and antiapoptotic protein inhibition via DNA damage and disrupting the mitochondrial membrane. Wypij et al. [13] showed that biosynthesized AgNPs from actinobacterial strain SF23 capped with proteins might be a potential cytotoxic agent against cancer cells and bacteria. Another study demonstrated that β-sitosterol-derived AgNPs can effectively induce cytotoxicity in human colon cancer cells through induction of early apoptosis via enhanced p53 protein expression [14]. However, the precise mechanism of AgNPs as antimicrobial and anticancer agents has not yet been elucidated [15]. AgNPs have wide-spectrum and highly antimicrobial and anticancer activity even at very low concentrations, so this study aimed to investigate the antibacterial and anticancer effects of fungus-mediated AgNPs when used singly and synergistically with biochar. Biochar is the biomass produced by heating in the absence or limited presence of air at temperatures above 250 °C, in a procedure called pyrolysis. Combinations of biochar and AgNPs could minimize the bioactive agents in the dosage, thereby lowering their noxiousness and boosting their potential antimicrobial and anticancer effects.

## 2. Results

### 2.1. Particle Size, Zeta Potential and Morphology

The average hydrodynamic diameter of AgNPs was found to be around 45 nm with a particle distribution index of 0.231, and they were found to be highly stable with a zeta potential of −19.6 mV (Figure 1A). According to zeta measurements of the particle size distribution, 97.3% of the AgNPs nanoparticles were 37.62 nm in size (Figure 1B). The examined particles are composed of a large number of tiny particles that are less than 0.5μm in size, as seen in the transmission electron microscope (STEM) images (Figure 2A,B). The scanning transmission electron microscopy micrograph of biochar combined with AgNPs is shown in Figure 2C.

### 2.2. XRD Analysis

The crystalline nature of the biosynthesized Ag nanoparticles was confirmed by the X-ray diffraction analysis (Figure 3). Two distinct diffraction peaks at 2θ values of 38.1°, 44.1° (shown with stars) can be assigned to the (1 1 1), (2 0 0) plane, respectively. The appearance of these patterns indicates that the formation of silver nanoparticles is fcc and crystalline in nature (JCPDS file no. 84-0713 and 04-0783). The appearance of several unassigned peaks that were observed in the XRD spectrum could be related to the crystallization of the bioorganic compounds from fungi during the preparation of the silver nanoparticles, which further confirms the formation of ligand interaction between the silver nanoparticles and the bio-organic compounds from the fungal extract. Many research groups have reported similar results during the preparation of Ag nanoparticles using edible mushroom extract and geranium leaves.

### 2.3. ATR-IR Spectra Analysis

The ATR-IR spectra of silver nanoparticles produced biologically (Figure 4) show numerous distinct peaks at 633, 989, 1110, 1663, 1704 and 2450, which demonstrated the presence of several organic functional groups that act as a reducing and stabilizing agent on the surfaces of silver nanoparticles. The analysis of this spectrum showed a very broad band around 3080 to 3437 cm^−1^ assigned to O–H and ~N–H stretching vibrations of amide. The presence of the main amine group of protein is shown by the brevity of the broad peak at 2450 cm^−1^. The appearance of several bands in the regions from 600–900 cm^−1^ corresponds to C-H aromatic out of the plane. The band at 1259 cm^−1^ could be related to the formation of C-O-C stretching in aromatic rings. The strong wide peak between 1400 to 1450 cm^−1^ corresponds to C-H stretching, while the weak band at around 1520 cm^−1^ represents the formation of C=C aromatic. Meanwhile, bands in the range of 2900–2990 cm^−1^ correspond to C-H stretching. The presence of the amide in the protein’s C=O stretching vibration is indicated by the peaks at 1704, 1663 and 1110 cm^−1^. The stretching frequencies of the amino and amino-methyl stretching groups of proteins have obvious peaks around 1340 cm^−1^. These organizations may be in charge of the synthesis and conformation of the biological system that accompanies the silver nanoparticles.

### 2.4. Antimicrobial Activity of Silver Nanoparticles

The potential interactions between AgNPs and biochar were evaluated using the disc diffusion method or MIC utilizing microdilution techniques (Table 1). When AgNPs were investigated alone, the inhibitory zones shown by AgNPs against *S. epidermidis*, *S. aureus*, *E. coli*, *P. aeruginosa*, *P. aeruginosa* ATCC 10,145 and *E. coli* ATCC 25,922 were 17.5, 18.5, 14.0, 12.5, 12.3 and 13.5 mm, respectively. However, none of the studied bacteria were inhibited by biochar. AgNPs and biochar together significantly improved (*p* < 0.05) the antibacterial activity of AgNPs against all bacterial species. When AgNPs and biochar were used together, the inhibiting zones that resulted were as follows: 19.5 ± 0.0 mm, 14.5 ± 0.4 mm, 21.5 ± 0.5 mm, 16.5 ± 0.0 mm, 16.5 ± 0.5 mm and 15.0 ± 0.6 mm. When using silver nanoparticles alone, the MIC values ranged between 6.38 and 19.15 µg/mL. However, when biochar was combined with silver nanoparticles, the range of MIC values decreased by approximately two-thirds, with a range of 2.13–6.38 μg/mL. The lowest MIC values obtained were for *S. epidermidis* and *S. aureus* (2.13 μg/mL). Although MIC and MBC values were discovered to differ for various bacteria when they were exposed to AgNPs biosynthesized by fungi, for each bacterium, the MBC matched the MIC exactly.

### 2.5. Modulation of Proliferation of HT29 Colorectal Cancer Cell Line as Well as Fibroblasts by Biochar and AgNPs

Figure 5a–d illustrate the antiproliferative abilities of biochar and AgNPs on the HT29 colorectal cancer cell line and fibroblast cell line. The toxicity effect of biochar on HT29 was relatively reduced in almost all concentrations that were tested, compared to the cytotoxicity effect of AgNPs on the same cell line. Furthermore, the cytotoxicity results demonstrated the lack of selectivity of AgNPs compared to biochar against normal fibroblasts.

### 2.6. The Combined Effects of Biochar and AgNPs

A concentration of 6 μg/mL of AgNPs was used in subsequent synergy experiments with biochar against the HT29 cell line. The antiproliferative efficacies of biochar with AgNPs on the HT29 cell line are further illustrated in Figure 6. Nevertheless, AgNPs lacked selective cytotoxicity in fibroblasts. However, the AgNPs combined with biochar had an equipotent effect or slightly better effect than AgNPs alone as well as had selective cytotoxicity in fibroblasts.

### 2.7. Inflammation and Angiogenesis

A concentration of 6 μg/mL of AgNPs was used in subsequent inflammation and angiogenesis experiments using the HT29 cell line. The anti-inflammatory activity of biochar with AgNPs on the HT29 cell line is shown in Figure 7. AgNPs, biochar and the combination of AgNPs and biochar showed a significant (*p* < 0.05) low expression level of TNF alpha, interleukin 6 and interleukin 1 beta. Furthermore, Figure 8 shows the significant (*p* < 0.05) downregulation of VEGF under the influence of both AgNPs and biochar.

## 3. Discussion

The average hydrodynamic size diameter of AgNPs was found to be around 45 nm with a particle distribution index of 0.231, and they were found to be highly stable with a zeta potential of −19.6 mV. A polydisperse distribution can be identified as such if the PDI (DLS) is greater than 0.2 [16]. Since these particles are electrostatically stable and hence resist self-assembly, they exhibit Z values of this size, which are typical of particles with a substantial amount of charge. The density distribution of our sample shows the degree of dispersion of various sizes. Given that particle size and light scattering are connected, a small portion of a larger component might dominate. A sample’s optical characteristics are used in the number distribution to display the relative number based on the size distribution obtained from data on the density distribution [17,18]. Size distribution of nanoparticles Zeta-measurements showed that 97.3% of the AgNPs are 37.62 nm in size. The examined particles are composed of a large number of tiny particles that are less than 0.5 μm in size, as seen in the scanning transmission electron microscopy (STEM) images. The results of the SEM measurement were examined using the open-source image-processing tool ImageJ [19]. AgNPs generated by biosynthesis were assumed to be spherical with a mean particle size of 30 ± 4.3 nm based on 100 particle size measurements. It was difficult to determine the structure of the discovered AgNPs because the image was unclear when viewed at higher magnification. This may have been caused by sample charging, the presence of nonconducting carbon stabilizers or nanoparticle aggregation into larger composites [20]. Nanoparticles’ SEM images show that they are spherical and range in size from 10 to 80 nm. AgNPs shrank during the drying process, as shown by SEM micrographs, making them smaller than those seen by DLS analysis. AgNPs had an average size of less than 45 nm. 

The XRD pattern and the presence of peaks verify the synthesis of AgNPs. Resolved diffraction peaks verified the crystalline character of the manufactured AgNPs [21]. Preparation of Ag nanoparticles using edible mushroom extract [22] and geranium leaves [23] yielded similar findings. The existence of numerous organic functional groups that operate as a reducing and stabilizing agent on the surfaces of silver nanoparticles was shown by peaks in the ATR-IR spectra of recently synthesized silver nanoparticles [24,25]. 

High pyrolysis temperatures have an impact on biochar sorption because they cause an increase in surface area, microporosity and hydrophobicity [26,27]. Biochar is well adapted for use in the adsorption process as soil additives in agricultural settings because it enhances the soil’s physical and chemical properties by enhancing its water-holding capacity, nutrient retention, surface area and water resistance [28].

Date seed (DS) biochar produced at 550 °C was found to be appropriate for remediation of metal-contaminated water. This was evident in the data, which showed that applying the biochar to *Raphanus* sp. and *Arabidopsis* sp. reduced metal stress and toxicity [29]. In this study, a combination of 6 μg/mL AgNPs with different concentrations of biochar was made to assess its activity against colorectal cancer cell lines as well as against pathogenic bacteria. Our results showed that biochar had no effect on the growth of either Gram-positive or Gram-negative bacteria, suggested that biochar may have an indirect antimicrobial effect, via altering some bacteria’s metabolic processes [28,30,31]. In contrast, AgNPs suppressed the growth of *S. epidermidis*, *S. aureus*, *E. coli*, *P. aeruginosa* and *P. aeruginosa* ATCC 10145. When using silver nanoparticles alone, the MIC values ranged between 6.38 and 19.15 μg/mL. However, when biochar was combined with silver nanoparticles, the range of MIC values decreased by approximately two-thirds, with a range of 2.13–6.38 μg/mL. Although MIC and MBC values were discovered to differ for various bacteria when they were exposed to AgNPs biosynthesized by fungi, for each bacterium, the MBC matched the MIC exactly, suggesting that antibacterial activity is strain(s) specific [32]. As AgNPs have a broad antibacterial range and significant antimicrobial activity, they can efficiently kill a variety of organisms even at extremely low concentrations [33]. Reactive oxygen species (ROS) generation, change in DNA structure and destruction of bacterial cell walls have all been extensively acknowledged as AgNP-related antimicrobial processes [7,34]. Rare occurrences of bacterial resistance to AgNPs have been observed, in contrast to the potential for antibiotic resistance, which might limit the applications of medical technology [35]. These consequences specify that silver nanoparticles could afford a safer alternative to conventional antimicrobial agents in the antimicrobial formulation [36]. Additionally, the results demonstrate that when AgNPs were incubated with biochar on colorectal cancer cells, their cytotoxicity activity increased at low concentrations but their toxicity to fibroblast cells was reduced.

Cell viability evaluations are crucial for bioassays that reveal how cells respond to toxic compounds since they may reveal details on metabolic activity, cell death and survival. These analyses demonstrate how cells respond to dangerous substances [37]. The AgNPs’ increased surface area caused them to interface with bacteria on their surface more frequently, that also enhanced their bactericidal activity. Free radicals were produced and DNA was oxidized as a consequence of the bacterial cell wall being punctured and then destroyed [38]. However, the combination of biochar and a sublethal amount of AgNPs (5 μg/disc) fairly quickly inhibited a wide variety of Gram-positive and Gram-negative bacteria. By combining biochar and silver nanoparticles, a novel potency of silver nanoparticles (AgNPs) with significantly improved antibacterial and therapeutic efficacy was created. The impact of mixed AgNPs (100–1000 mg/mL) and biochar (2% *w*/*v*) on maize seedlings in a hydroponic exposure medium was previously studied by Abbas et al. [39,40]. The concentration of Ag+ ions in the growth medium dropped as a result of the complexing of the biochar surface brought on by the interaction between the AgNPs and the biochar. Subsequently, the bioavailability of the AgNPs was reduced. The addition of biochar decreased the phytotoxicity of the AgNPs by a ratio of four to eight. The reduced oxidative stress in plants treated with AgNPs and biochar was also responsible for the increased activities of antioxidant enzymes. Based on our findings, biochar made from date seeds is an efficient tool for decreasing AgNPs’ bioavailability. This reduces the toxicity of the AgNPs while limiting their release, resulting in greater selectivity for colorectal cells over fibroblasts. When silver nanoparticles (AgNPs) come into contact with biochar, they likely become immobilized and lose some of their bioavailability. The increased surface area of the AgNPs combined with the complexing of the biochar surface that was caused by the interaction with the AgNPs resulted in the creation of a novel potency of silver nanoparticles (AgNPs) that substantially enhanced antibacterial and therapeutic efficacy. 

According to research that agrees with our findings, AgNPs may have antitumor effects by inhibiting cell proliferation and inducing proapoptotic events through the p53, Bax/BCL-2 and caspase pathways [11], as well as by causing DNA fragmentation and altering cellular redox status in cancer models for cervical, breast, lung, nasopharyngeal, hepatocellular, glioblastoma and colorectal cancer [9]. 

In addition, it was discovered that colon cancer cell lines (HCT-116, HT29 and SW620) were sensitive to the anticancer effects of green-synthesized AgNPs made from *Commiphora gileadensis* stem extracts. Four genes (CHEK1, CHEK2, ATR and ATM) were measured to determine the anticancer activity of green-synthesized AgNPs using real-time polymerase chain reaction (RT-PCR) [41]. In a recent study, the ability to treat colorectal cancer (HT29), colorectal carcinoma (HCT-116), ileocecal colorectal adenocarcinoma (HCT-8 (HRT-18)) and Burkitt’s lymphoma (Ramos.2G6.4C10) cell lines was demonstrated via a combination of graphene oxide (GO) and silver nanoparticles. According to this study, the combination of GO and AgNPs holds significant promise as a novel class of chemotherapeutic agents for the treatment of colon cancer [42]. Additionally, it has been demonstrated that, compared to 5 FU, the biogenic AgNPs produced using honeybee extract exhibited anticolon cancer activities at the cellular and molecular levels [43]. The ionic Ag+ species, which can enter cells and affect nucleic acids (such as starting DNA condensation), form complexes with electron donor groups that are required for protein function and affect cell signaling cascades, is particularly prone to be released by AgNPs [7]. Therefore, nanosilver is a prime option for use in clinical and therapeutic applications as it is less reactive than silver ions [44].

The administration of AgNPs by employing the HT29 cell line as a model results in a decrease in all proinflammatory cytokines examined, including IL-1 beta, IL-6 and TNF-α. Our results are consistent with information from several studies that provide comparable results with AgNPs created using plant extracts, indicating that they have some anti-inflammatory potential [45,46,47]. However, in other study, mammalian macrophages were employed to investigate the direct effects of biochar on immune cells, and the findings revealed that biochar immediately lowered the levels of inflammatory cytokines produced by in vitro activated macrophages [30]. As a result, there is a clear connection between AgNPs and biochar due to the fact that both substances have anti-inflammatory properties on their own, and according to our research, the combination of AgNPs and biochar has anticancer effects. Many chronic inflammatory ailments can be treated effectively by therapeutic targeting of TNF-α [48,49]. Inflammation and cancer appear to be linked, and TNF-α appears to be a key intermediary in that relationship [50]. We infer from this association that the combination of AgNPs and biochar has a true synergistic effect pertaining to cytotoxicity on HT29 cells. Certain bioactive chemicals have been shown in other studies to have an effect on many antiapoptotic genes, hence preventing these genes from performing their early protective mechanism against apoptosis and eventually causing the death of apoptotic cells [51,52,53]. 

Additionally, it was proposed that AgNPs could control the activation of the TNFR1/NF-KB transcriptional pathway, leading to a considerable decrease in the proinflammatory cytokines in a lung epithelial cell line [54]. In addition, the current study investigated the antiangiogenic activities of AgNPs by using a colorectal cancer cell line as a model for this experiment. The findings suggest that AgNPs with biochar could significantly reduce the expression level of VEGF when compared to untreated cells [55,56,57,58,59]. We hypothesized that the combination of biochar and AgNPs could have an effect on the biological processes that are responsible for neovascularization and inflammation. Studies examining the effects of the newly combined biosynthesised AgNPs and biochar will undoubtedly aid in elucidating the specific mechanism of antitumor action of a potential combined cancer therapy. In conclusion, biochar might prove to be a potential nanocarrier for treating pathogenic multidrug-resistant bacteria and enhancing treatment by joining with AgNPs or other chemotherapeutic agents.

## 4. Materials and Methods 

### 4.1. Fungal Strain

The fungal strain known as W7B was extracted from soil samples collected in the Al-Karak district of southern Jordan. The ITS sequencing allowed for the determination of the species of the fungal strain (Macrogen, Seoul, Korea). The NCBI database was used to perform a sequence similarity comparison, which indicated that the ITS sequence of the fungus W7B was 100% identical to the ITS sequence of *Emericella dentata*. The accession number for Emericella dentata was MH032749.

### 4.2. Bacterial Strains and Reagents

All of the compounds and media that were utilized were sourced from Sigma-Aldrich. Both Gram-positive and Gram-negative organisms were utilized in the examination of the antibacterial properties. Karak Government Hospital supplied all of these isolates (KGH). These isolates were identified, and their antibiotic profiles were determined, using the BIOMERIEUX VITEK® 2 SYSTEM. Each bacterial strain was subcultured twice on nutritional agar at 37 °C for 24 h prior to testing to guarantee viability, and bacterial cultures were maintained in a refrigerator at 4 °C for subsequent use. All of the solutions were made with highly purified deionized water. Pyrolysis at 550 °C resulted in the production of biochar, as had been documented previously [29].

### 4.3. Isolation and Screening of AgNP-Producing Fungi

The ability of the fungal strain to generate AgNPs was looked into using a previously described method [60]. In summary, an *Emericella dentata* fungal isolate was initially cultivated aerobically at 30 °C for a number of days in the proper conditions. The cells were suspended in the same sterile medium and centrifuged at 10,000 rpm for 20 min to produce a fungal extract, and the supernatant was combined in a 1:1 ratio with 1 mM silver nitrate. Then, the pH of the reaction mixture was adjusted to 8.5. The obtained mixtures were shaken (200 rpm) at 37 °C in the dark until a transition from a pale yellow to a dark brown color occurred.

### 4.4. AgNP Characterization

The absorbance of the produced brown color, which denoted the formation of AgNPs, was determined spectrophotometrically using a UV-1800 spectrophotometer. The reaction solution was repeatedly centrifuged at 15,000 rpm for 20 min to remove the bioformed silver nanoparticles (MIKRO 200 R, Hettich, Germany). The nanoparticles were then resuspended in deionized water for washing. Following recovery, the pellet of silver nanoparticles was vacuum dried (VWR 1410 Vacuum Oven, USA). Using an FEI Versa 3D Dual Beam apparatus (FEI, USA), STEM pictures of the produced AgNPs’ size, distribution and form were obtained. The crystalline phase of the created AgNPs was identified using the MAXima-X XRD-7000. To determine the proteins and other functional groups that contribute to AgNP stability, a Bruker Alpha FTIR spectrometer was used [16]. 

### 4.5. Nanoparticle Zeta Potential and Size Distribution

The Zetasizer Nano ZS90 was utilized in order to ascertain not only the zeta potential of the AgNPs but also the particulate size of the AgNPs (Malvern Instruments, Malvern, UK). The investigation was carried out at a temperature of 25 °C and a dispersion angle of 90° using samples of varying concentrations that had been diluted with deionized and distilled water, respectively.

### 4.6. Antibacterial Activity of Silver Nanoparticles, Biochar and Their Combination

The biosynthesized AgNPs, biochar and their combination were evaluated against Gram-positive bacteria, namely *S. epidermidis* and *S. aureus*, and Gram-negative bacteria *P. aeruginosa*, *E. coli*, *P. aeruginosa* (ATCC 10145) and *E. coli* (ATCC 25922). The pathogenic strains were identified using the BIOMÉRIEUX VITEK^®^ 2 system or Enterosystem 18 R (Liofilchem) after being isolated from UTI patients. The ATCC strains were taken from a lab stock as pure cultures. The antibacterial activities of AgNPs and biochar were evaluated using the disc diffusion method [61]. Briefly, 250 mL of bacterial suspension adjusted to 10^6^ was mixed with 30 mL of melted Mueller–Hinton agar. After solidification, a sterilized disc (6 mm) containing AgNPs (10 μg/disc), biochar (10 μg/disc) or their combination (5 μg/disc of AgNPs and 5μg/disc of biochar), or negative control (DMSO), was transferred aseptically to the surface of the inoculated agar. Then, the plates were incubated at 37 °C for 24 h and the inhibition zone diameter was measured in millimeters. Each sample was tested in triplicate. The stock solution for AgNPs, biochar and their combination was prepared as follows: 1 mg/mL AgNP solution (AgNPs-S), and 1 mg/mL biochar solution (BS). Before use, BS was kept for 24 h at 25 °C while being shaken. The phosphate-buffered water used as the aqueous solution had a pH of 7.4 (Invitrogen, USA). AgNPs-S-BS solution was made by combining 100 μL of AgNPs-S and 100 μL of BS, which was vortexed multiple times before use. To obtain 10 μg/disc of AgNPs and 10 μg/disc of biochar, as well as a combined 5 μg/disc of biochar and 5 μg/disc of AgNPs, 10 μL each of AgNPs-S, BS and AgNPs-S-BS were then poured onto sterile 6 mm discs that had been put on a plate. A center disc, without AgNPs, biochar or their combination, was maintained as control. After that, the plates were kept for 24 h in a 37 °C incubator. The presence of growth-inhibitory zones was checked for after incubating the plates. The sizes of the zones of inhibition (ZOIs) were reported after being measured with a ruler. The test was repeated three times [62,63].

### 4.7. MIC Determination

The broth microdilution technique, recommended by the Clinical Laboratory Standards Institute, was used to determine minimum inhibitory concentrations (MIC) [64]. The test was run in duplicate in 96-well microtiter plates. Triplicate tests were run in 96-well microtiter plates containing nutrient broth for bacterial development. A maximum bacterial concentration of 5 × 10^5^ cfu mL^−1^ was maintained in each well of the microtiter plate. The final content of the manufactured AgNPs varied from 0.016 to 1024 g mL^−1^. There were two sets of standards maintained: a positive (broth containing a bacterial inoculum) and a negative (sterile, noninoculated) set. The abundance of the microbial inoculum was determined by counting the number of colonies. The soup was diluted (1:1000) with the microbial inoculum (5 × 10^5^ cfu mL^−1^), and then 100 μL was spread across the surface of Mueller–Hinton agar. The inoculum density was calculated to be 5 × 10^5^ cfu mL^−1^ if there were 50 colonies after incubation. The experiment involved a 24 h incubation period at 37 °C for the bacterial microtiter plates. Finally, the MIC values were manually estimated [13]. The minimum bactericidal concentrations (BCs) of AgNPs against bacterial isolates were also determined, which are defined as the lowest concentrations of antibacterial agents that inhibited the survival of >99.9% of bacteria. Each test sample was incubated for 100 μL before being spread onto an antibacterial agent-free medium. Bacterial growth was monitored on plates after a 24 h incubation period at 37 °C.

### 4.8. Cancer Cell Line Culture

A human colorectal cancer cell line, namely, HT29, and human periodontal ligament (PDL) fibroblasts were used. These cells were cultured in DMEM containing 10% FBS, HEPES buffer (10 mM), L-glutamine (100 μg/mL), gentamicin (50 μg/mL), penicillin (100 μg/mL) and streptomycin (100 mg/mL).

### 4.9. Cell Harvesting and Counting

All cell cultures were maintained at 37 °C in a humidified 5% CO_2_ environment. After rinsing the cells in 75 cm^2^ flasks with 3–5 mL of phosphate-buffered saline (PBS), 1–2 mL of trypsin was added to each flask, and incubation continued until the cells separated. Each cell line had the same volume of new media added to it, and then it was pipetted gently to break up any clumps and create a uniform single cell suspension. Each cell type had a different rate and proportion of proliferating cells. Once the desired quantity of cells was reached, the process of cell propagation was repeated every 2–3 days. Cells were enumerated by transferring a 25 μL suspension of harvested cells and 100 μL of trypan blue dye (4% final concentration) to the rim of a hemacytometer counting chamber [38].

### 4.10. Antiproliferative Activity of Silver Nanoparticles, Biochar and Their Combination

The biochar and AgNPs were assayed for cell toxicity. Cytotoxicity measurements were based on the viability of the cells present in the culture. Cells were seeded into 96-well plates at a density of 1 × 10^4^ cells per well and incubated for 24 h at 37 °C in DMEM, then incubated with DMEM containing biochar at different concentrations (6–200 μg/mL) and AgNPs (6–200 μg/mL) for 48 h. The subsequent procedure involved conducting the MTT assay, which utilizes 3-(4,5-dimethylthiazol-2-yl)-2,5-diphenyl-2H-tetrazolium bromide as follows: after removing the medium, the cells were cultured at 37 °C for 4 h with 20 μL of MTT solution (5 mg/mL). Next, 200 μL of DMSO was added was added to each well. The microtitre plate was placed on a shaker in order to dissolve the insoluble formazan crystals. We checked the optical intensity at two different wavelengths, 570 nm and 630 nm. The results were collected from three separate wells for accuracy. Primary cultures of human gingival ligament fibroblasts were tested for their ability to selectively inhibit proliferation using an IC50 value that is among the lowest reported. Antiproliferative activities were determined using triplicate assays, and the means (*n* = 3) were given ± SD [65,66,67,68].

To assess the synergistic effects of the biochar in combination with AgNPs, cells were seeded into 96-well plates at a density of 1× 10 ^4^ cells per well and incubated for 24 h at 37 °C in DMEM. After incubation, all cell lines were treated with a combination of biochar at different concentrations (6–200 μg/mL) and AgNPs at a concentration of 6 μg/mL. After 48 h, the MTT solution was added to each well and incubated for 4 h. The MTT–formazan crystals formed were dissolved in 100 μL of DMSO and the absorbance was measured at 570 nm and 630 nm. Data were obtained from triplicate wells. 

### 4.11. Inflammation and Angiogenesis Assays 

HT29 cancer cells were cultured in 6-well plates at a density of 500,000 cell/well until 80% confluent. On the day of the experiment, the cells were supplemented with AgNPs at 6 μg/mL and biochar at 200 μg/mL. After 72 h of incubation at 37 °C, incubation medium was collected and stored at −20 °C for a subsequent ELISA determination of the amount of secreted tumor necrosis factor alpha (TNF alpha), interleukin 6 (IL-6), interleukin 1 beta (IL-1 beta) and VEGF.

### 4.12. Statistical Analysis

Statistical differences between control and different treatment groups were determined using GraphPad Prism ANOVA followed by Dunnett’s post hoc test. For all statistical analyses, a *p*-value of less than 0.05 was considered statistically significant. *p*-values of less than 0.001 were considered to show a highly statistically significant difference.

## 5. Conclusions

This study represents the first investigation into the use of AgNPs and biochar in conjunction with one another to combat pathogenic microbes and colorectal cancer cells. Even though biochar by itself did not show any antibacterial activity, the combination of the two had antibacterial activity that was synergistic against all of the strains that were evaluated. Both colorectal and fibroblast cell lines, which were used in this study, were susceptible to AgNPs when they were used individually and even more so when they were combined with biochar. According to the findings of the current research, silver nanoparticles generated by fungi and biochar possess cytotoxic characteristics that can be utilized in various medical applications. Our research has shown that even minute concentrations of biosynthesized AgNPs, when combined with biochar, have the potential to treat drug-resistant bacteria as well as colorectal cancer epithelial cells. In spite of the fact that recent findings indicate that the combination of AgNPs and biochar might be an effective new therapeutic agent for the treatment of cancer, more research is needed to assist in the development of these agents so that they can be used in clinical trials.

## Figures and Tables

**Figure 1 antibiotics-12-00597-f001:**
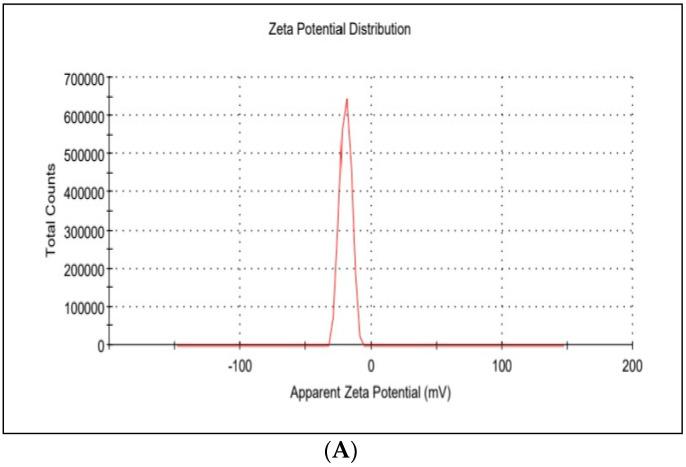
(**A**) Zeta potential distribution of silver nanoparticles. AgNP zeta potential is −19.6 mV; (**B**) the size distribution by the intensity of AgNPs.

**Figure 2 antibiotics-12-00597-f002:**
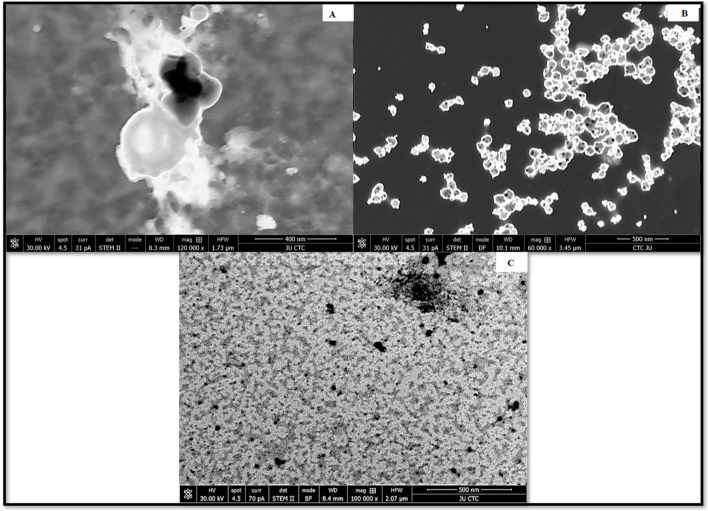
STEM micrograph of (**A**) biochar, (**B**) silver nanoparticles synthesized by the reaction of 1.0 mM silver nitrate with *Emericella dentata* filtrate and (**C**) combined AgNPs and biochar.

**Figure 3 antibiotics-12-00597-f003:**
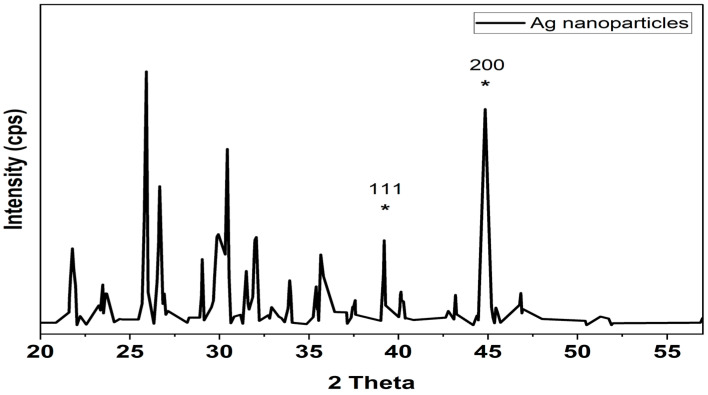
XRD analysis of silver nanoparticles biologically prepared with *Emericella dentata* filtrate. * Peaks are attributed to silver’s (1 1 1) and (2 0 0) planes of diffraction. The (1 1 1) and (2 0 0) diffraction peaks in an XRD pattern can be indexed.

**Figure 4 antibiotics-12-00597-f004:**
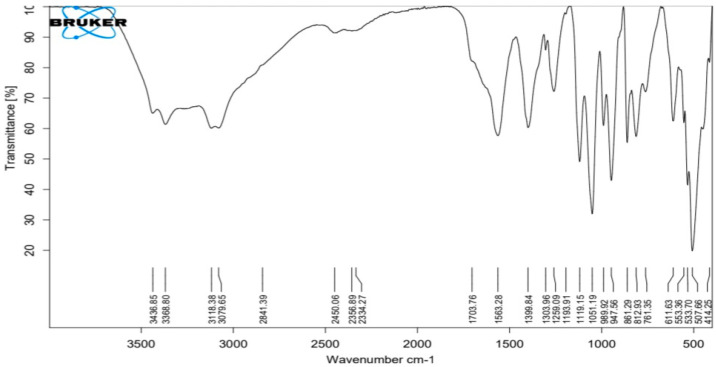
ATR−FTIR of silver nanoparticles biologically prepared with *Emericella dentata* filtrate.

**Figure 5 antibiotics-12-00597-f005:**
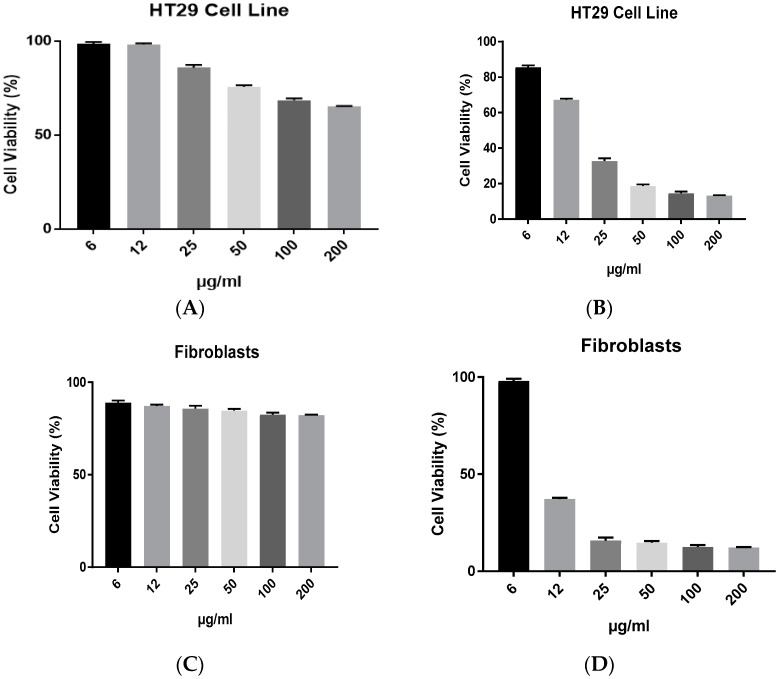
The effect of (**A**) biochar and (**B**) AgNPs on the viability of HT29 cell line and (**C**) biochar, (**D**) AgNPs on the viability of fibroblasts in 72 h cultures as measured by an MTT assay. The results are expressed as means of three measurements ± SD (*n* = 3–4 independent replicates).

**Figure 6 antibiotics-12-00597-f006:**
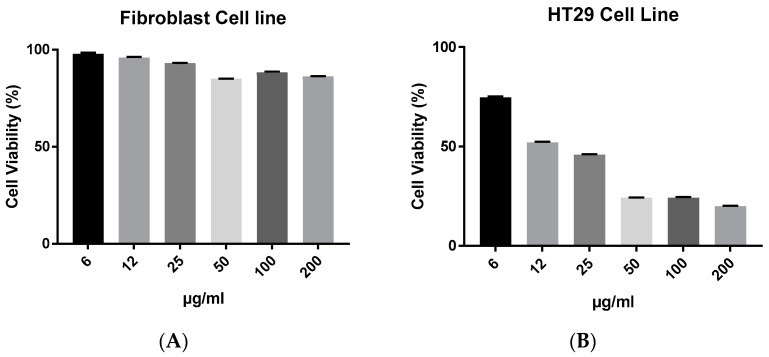
The effect of AgNPs at 6 μg/mL with biochar at 6 to 200 μg/mL on the viability of (**A**) fibroblasts and (**B**) HT29 cell line in 72 h cultures as measured by an MTT assay. The results are expressed as means of three measurements ± SD (*n* = 3–4 independent replicates).

**Figure 7 antibiotics-12-00597-f007:**
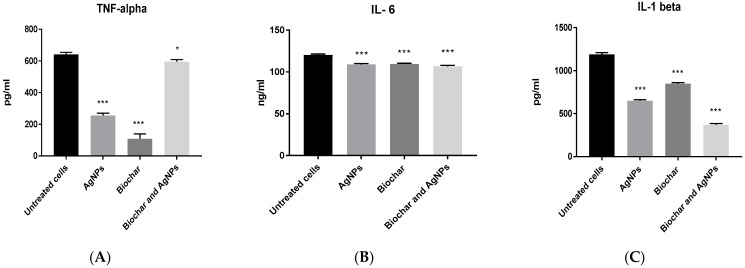
The effect of AgNPs at 6 μg/mL and biochar at 200 μg/mL on the expression of (**A**) TNF alpha, (**B**) IL-6 and (**C**) IL-1 beta using HT29 cell line in 72 h cultures. The results are expressed as means of three measurements ± SD (*n* = 3–4 independent replicates), *: *p* < 0.05, ***: *p* < 0.001 compared to respective untreated cells.

**Figure 8 antibiotics-12-00597-f008:**
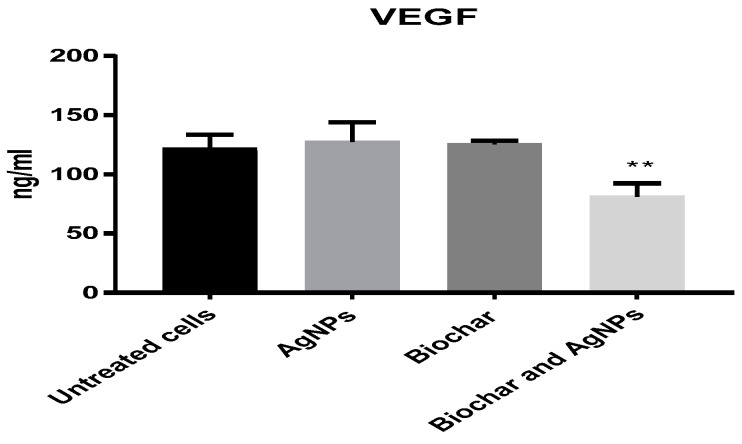
The effect of AgNPs at 6 μg/mL and biochar at 200 μg/mL on the expression of VEGF in 72 h cultures. The results are expressed as means of three measurements ± SD (*n* = 3–4 independent replicates). **: *p* < 0.01 compared to respective untreated cells.

**Table 1 antibiotics-12-00597-t001:** Inhibition zones and MIC values of AgNPs, biochar and AgNPs + biochar.

	AgNPs		Biochar		AgNPs + Biochar		*p*-Value between Inhibition Zone
Micro-Organism	Inhibition Zone (mm)	MIC (μg/mL)	MIC (μg/mL)	MIC (μg/mL)	Inhibition zone (mm)	MIC (μg/mL)	
*S. epidermidis*	17.5 + 0.7	6.38	0.0	Nd	19.5 + 0.0	2.13	0.0039 **
*P. aeruginosa*	12.5 ± 0.5	19.15	0.0	Nd	14.5 ± 0.4	6.38	0.0028 **
*S. aureus*	18.5 ± 0.5	6.38	0.0	Nd	21.5 ± 0.5	2.13	0.0009 ***
*E. coli*	14.0 ± 0.0	19.15	0.0	Nd	16.5 ± 0.0	6.38	<0.0001 ****
*P. aeruginosa* ATCC 10145	12.3.0 ± 0.0	19.15	0.0	Nd	16.5 ± 0.5	6.38	<0.0001 ****
*E. coli* ATCC 25922	13.5 ± 0.6	19.15	0.0	Nd	15.0 ± 0.6	6.38	0.0188 *

Nd: not detected. The results are expressed as means ± SD (*n* = 3–4 independent replicates). *: *p* < 0.05, **: *p* < 0.01, ***: *p* < 0.001, ****: *p* < 0.0001 compared to AgNPs + biochar sample with respective AgNP samples.

## Data Availability

The datasets used and analyzed during the current study are available from the corresponding author upon reasonable request.

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
