# Peer review of "Fungal-Mediated Silver Nanoparticle and Biochar Synergy against Colorectal Cancer Cells and Pathogenic Bacteria"

_antibiotics, 2023, doi:10.3390/antibiotics12030597_

Round 1
Reviewer 1 Report
REVIEW
The article intitled “Fungal-mediated Silver Nanoparticle and Biochar Synergy against Colorectal Cancer Cells and Pathogenic Bacteria” evaluated AgNPs - developed by green synthesis using fungi - antibacterial and anticancer properties alone and combined with Biochar. The modulatory effect of AgNPs on the Biochar effect toward cancer cells and the reversion effect of that modulation on TNF-alfa are particularly interesting results that must be discussed deeply.
Below are listed some suggestions and considerations regarding the manuscript:
INTRODUCTION AND METHODS
§ The name of all species of bacteria and fungi must be written in italic throughout the text.
§ In line 60 – I suggest insert: fungal-mediated AgNPs.
§ Between lines 248 and 249 is there some repeated information about centrifugation? Please, clarify this process.
§ How did you impregnate the tested agents (AgNPs and Biochar) in the disc? Did you make a solution of agents using DMSO? How many microliters did you inserted on the disc to achieve 10 micrograms (or 5/5)? How did you dry the discs?
§ Please, correct superscript in lines 241, 279, 292, 293, 296, 298, 336.
§ In 4.7, the MBC test must be clarified.
§ The evaluated concentrations of tested agents must be described in materials and methods section (lines 325, 337, 344, 345).
§ I miss the control agents in inflammation and angiogenesis assays.
RESULTS AND DISCUSSION
§ Could the images be enlarged?
§ Figure 2(c) must be explored.
§ Uv-vis, XRD and FTIR results must be presented for AgNPs characterization.
§ Please explain: “a definite inhibitory of up to 70%” (lines 101 and 102).
§ Please correct Table 1 (Biochar Inhibition – fourth column).
§ I suggest using T-test o compare the results of inhibition zones between AgNPs and AgNPs+Biochar. In results, the authors observe difference, but we need to confirm statistically.
§ Please correct Figure 3 legend. The graphs do not point the modulatory results.
§ Please, some interesting results deserve a deeper discussion:
Ø How may you explain the modulatory effect of Biochar on AgNPs antimicrobial effectiveness against tested bacteria? (Table 1)
Ø How may you explain the very interesting modulatory effect AgNPs (6 µg/mL) on Biochar cytotoxicity on HT29? (Figure 4 – please insert letters A and B/ Figure 5, A, B and C)
Ø Please, check and explain the effect and importance of TNF-alpha, IL-6, IL-1beta and VEGF on colorectal cancer management.
Ø How may you explain the very interesting modulatory effect (reversion effect) presented in Fig 5A? The interaction of AgNPs and Biochar promoted only a discreet difference to the control (effect on TNF-alpha levels) in comparison to the isolated effects. Could be this effect good or bad to colorectal cancer treatment!?
§ In general, I miss a deeper discussion of the biological tests results.
§ Please, correct the information in line 173. In fact, you have evaluated a combination of 6 µg/mL AgNPs, with different concentrations of Biochar.
§ I suggest rewrite the conclusion based on your interesting results.
§ Why did you concluded that: “Additional research into the anti-inflammatory and anti-angiogenic effects of the combination of AgNPs and biochar has uncovered the probable mechanisms that are involved in the activation of the TNFR1/NF-KB transcriptional pathway as a result of the regulation of these effects?
Kind Regards.
Reviewer 2 Report
The way in which the particle size is described is confusing: in line 85 it is said that 97.3% of nanoparticles have a size of 37.62 nm and that the size range goes from 37 to 175 nm and that the average size is of 175.3 nm.
This is all confusing, since TEM images are said to be presented when in fact they are SEM images.
It is described in the study that various equipment was used, such as: MAXima-X XRD-7000, Bruker Alpha FTIR spectrometer, etc. but the graphs are not shown and nothing is discussed. Is there a limited number of figures?
I consider that the most important thing in the article is not discussed:
1.- If the biochar is obtained by pyrolysis at 250 degrees Celsius, what may be affecting the selective cytotoxicity of fibroblasts?
2.- What is it that generates a synergistic effect between the silver nanoparticles and the biochar so that the diameter of the inhibitory halos increases and at the same time decreases the minimum inhibitory concentration (MIC)
Reviewer 3 Report
Hello Dear colleague, your manuscript is very interesting and here are some recommendations that may help to improve the paper
Kindly review the punctuation throughout your abstract.
Line 99, insert the value of the MIC after the words “ …had a MIC of’’ and remove the values of the interval in brackets.
Line 170, properly write 550°C.
Lines 264-265: Where are the results if FTIR in your results part? Kindly add a figure about
Line 299, complete the sentence after the words ‘’…..after being’’.
Line 307, complete the sentence.
Line 312, CO2 should be correctly written. Not CO2.
Line 309: L- glutamine (100 ìg/ml). What does ìg/ml mean? Kindly check and correct it
Line 324: Improve the way you wrote 1× 10 4 cells;
Round 2
Reviewer 1 Report
Suggestions have been accepted and improved the quality of manuscript.
Author Response
Thanks a lot to the reviewer for providing valuable comments to improve our manuscript.
Reviewer 2 Report
In the line 72 says "The examined particles are composed of a large number of tiny particles that are less than 0.5μm in size, as seen in the transmission electron micro scope (TEM) images (Figures 2a and b) and in lines 120 to 122 Figure 1. (A) Zeta potential distribution of silver nanoparticles. AgNPs zeta potential shows - 19.6 mV; (B) The size 120 distribution by the intensity of AgNPs nanoparticles. The size of nanoparticles ranges from 37.62 to 175.3 nm, and the average size is about 175.3 nm.
Based on this, the following observations were made:
1.-The images presented are not TEM, according to the manufacturer's description for the model described and used in this research work, the microscope is a -High-resolution field emission SEM- (FESEM) (https://www .microscop.ru/uploads/VERSA3D.pdf)
2.- If 97.3% of the nanoparticles have a size of 37.62 nm and their size range is 37.62 - 175.3 nm, the average size cannot be 175.3 nm if you also consider that there are a large number of particles smaller than 500 nm the average cannot be the one reported and apparently these nanoparticles are not being considered
Was the measurement of the nanoparticles observed by electron microscopy performed? how many?
What was the method to determine the average size of the nanoparticles and to obtain the value of 175.3 nm?
Place the number of counts (cps) on the "y" axis of the X-ray diffraction pattern
On line 200 of write ). ). *: this is correct? I think there is a parenthesis and a period left over
Author Response
Thanks a lot to the reviewers for providing valuable comments to improve our manuscript. We have corrected all the necessary comments.
